# Recent Advances in Drug Development for Hair Loss

**DOI:** 10.3390/ijms26083461

**Published:** 2025-04-08

**Authors:** Jino Kim, Seung-Yong Song, Jong-Hyuk Sung

**Affiliations:** 1New Hair Institute, Seoul 06034, Republic of Korea; jino@inewhair.com; 2Institute for Human Tissue Restoration, Department of Plastic and Reconstructive Surgery, Yonsei University College of Medicine, Seoul 06134, Republic of Korea; PSSYSONG@yuhs.ac; 3Epi Biotech Co., Ltd., Incheon 21983, Republic of Korea

**Keywords:** hair loss, drug development, stem cell therapy, antibody therapy

## Abstract

Hair loss disorders pose a substantial global health burden, affecting millions of individuals and significantly impacting quality of life. Despite the widespread use of approved therapeutics like minoxidil and finasteride, their clinical efficacy remains limited. These challenges underscore the pressing need for more targeted and effective therapeutic solutions. This review examines the latest innovations in hair loss drug discovery, with a focus on small-molecule inhibitors, biologics, and stem cell-based therapies. By integrating insights from molecular mechanisms and leveraging advancements in research methods, the development of next-generation therapeutics holds the potential to transform the clinical management of hair loss disorders. Future drug development for hair loss disorders should prioritize antibody therapy and cell-based treatments, as these approaches offer unprecedented opportunities to address the limitations of existing options. Antibody therapies enable precise targeting of key molecular pathways involved in hair follicle regulation, providing highly specific and effective interventions. Similarly, cell-based therapies, including stem cell transplantation and dermal papilla cell regeneration, directly address the regenerative capacity of hair follicles, offering transformative potential for hair restoration.

## 1. Introduction

Hair loss disorders, including androgenetic alopecia (AGA) and alopecia areata (AA), are among the most prevalent dermatological conditions affecting millions of individuals worldwide. These disorders can result from genetic predisposition, autoimmune reactions, hormonal imbalances, environmental stressors, and aging [1,2,3,4]. While hair loss is often considered a cosmetic issue, it can significantly impact self-esteem, mental health, and overall quality of life. Despite the growing demand for effective treatments, current therapeutic options remain limited in efficacy and scope.

Existing pharmacological treatments, such as minoxidil (a potassium channel opener) and finasteride (a 5α-reductase inhibitor), have demonstrated partial success in slowing hair loss and promoting regrowth. However, their effects are often temporary, and many patients experience inadequate responses or undesirable side effects. In recent years, advancements in molecular biology, regenerative medicine, and targeted drug development have paved the way for novel therapeutic strategies. Understanding the key molecular pathways that regulate hair follicle cycling, stem cell activity, and immune responses is crucial for developing more effective and personalized treatments for hair loss disorders.

This paper explores the latest approaches in hair loss drug development, focusing on emerging small-molecule inhibitors, biologics, and stem cell-based therapies (Figure 1). It also examines the role of androgen receptor (AR) and critical signaling pathways such as Wnt/β-catenin, JAK/STAT, and CXCL12/CXCR4 in hair follicle regulation. By integrating insights from molecular dermatology and advanced drug discovery techniques, we aim to provide a comprehensive overview of the current landscape and future prospects in hair loss treatment.

## 2. Current Approved Hair Loss Treatments

Several pharmacological treatments are currently available for hair loss disorders, with minoxidil, finasteride/dutasteride, and JAK inhibitors being among the most widely used (Table 1). While these medications have shown varying degrees of efficacy, they also have limitations that hinder their widespread success.

### 2.1. Minoxidil

Minoxidil is a potassium channel opener that promotes vasodilation, improving blood flow to hair follicles [5,6]. It extends the anagen phase of the hair cycle and stimulates dermal papilla cell (DPC) activity [7]. Approved by the FDA in 1988 for both male and female AGA, minoxidil is available as a topical solution or foam. However, its exact molecular mechanism remains only partially understood. Hair regrowth usually becomes visible after 3–6 months of continuous use, with peak efficacy at around 12 months. Dawber and Rundegren reported that the affected area became smaller in 561 of the 904 eligible subjects (62.0%), remained unchanged in 317 subjects (35.1%), and became larger in 26 subjects (2.9%) [8]. Hair density increases by approximately 10–30%, while hair thickness improves by 10–25% with Minoxidil treatment [9,10]. A 5% minoxidil solution is more effective than the 2% formulation, but women are generally advised to use the 2% solution due to potential side effects. Additionally, discontinuation leads to resumed hair loss. Common side effects include scalp irritation, itching, and unwanted facial or body hair growth (hypertrichosis). Low-dose oral minoxidil (0.25–5 mg/day) is emerging as an alternative to topical application, and studies suggest that it offers better patient adherence than the topical form [11,12]. However, side effects such as low blood pressure, swelling, and excessive hair growth in unwanted areas have been reported.

### 2.2. 5a-Reductase Inhibitors

Hair follicles in the scalp (especially the front and crown areas) exhibit high levels of AR expression. Testosterone and DHT bind to these receptors, triggering molecular signals that shorten the hair growth cycle. DHT shortens the anagen phase, meaning that hair stops growing sooner and falls out more quickly. Over time, hair follicles produce only vellus hairs, leading to baldness. Additionally, DHT increases inflammation and promotes fibrosis (scar tissue formation) in hair follicles, which makes it harder for follicles to recover, accelerating permanent hair loss.

Finasteride and dutasteride inhibit 5α-reductase enzymes, blocking the conversion of testosterone to dihydrotestosterone (DHT)—a crucial mediator in hair follicle miniaturization. By reducing scalp DHT levels, these drugs can slow hair loss progression and, in some cases, promote regrowth. Recent findings indicate that 5α-reductase type 1 and 3 (SRD5A1 and SRD5A3) are predominantly expressed in hair keratinocytes, and that DHT activation of the AR in DPCs triggers hair miniaturization [13].

Finasteride was FDA-approved for male AGA in 1997 (although not approved for women due to potential side effects) and is administered orally. Male AGA patients experience significant hair regrowth, but finasteride is ineffective for autoimmune-related hair loss. Finasteride increased hair density by 10–20% in cases of AGA after one year, and thicker hair shafts were observed in most patients. Over 80% of men maintained their existing hair over a five-year period. Finasteride is more effective than minoxidil for preventing hair loss. Some studies suggest that combining finasteride and minoxidil provides better results than using either treatment alone [14]. A 1 mg dose of finasteride is optimal for treating male pattern baldness, as 5 mg does not significantly increase hair growth but raises the risk of side effects.

Side effects such as erectile dysfunction, reduced libido, and gynecomastia occur in some men and may persist even after stopping therapy [15,16]. A systematic review suggested that dutasteride (0.5 mg) was more effective than finasteride (1 mg) in increasing hair counts (up to 1.5-fold), but approved in Asian countries. Finasteride inhibits only Type II 5α-reductase, while dutasteride inhibits both Type I and Type II 5α-reductase, making it more potent in reducing DHT levels [17]. Due to its longer half-life, dutasteride provides sustained benefits but also carries a higher risk of side effects [18].

### 2.3. JAK Inhibitors

Alopecia areata is an autoimmune disorder in which the immune system mistakenly attacks hair follicles, leading to sudden hair loss. This condition typically causes round or oval bald patches on the scalp, but in severe cases, it can result in total scalp hair loss (alopecia totalis) or complete body hair loss (alopecia universalis).

JAK inhibitors work by blocking the activity of Janus kinase enzymes, thereby interrupting the inflammatory signaling pathways that contribute to hair follicle miniaturization and loss. By modulating these pathways, JAK inhibitors can promote hair regrowth and extend the anagen phase of the hair cycle [19,20].

Early studies demonstrated that these JAK inhibitors could effectively reverse hair loss in patients with moderate to severe AA. This oral JAK inhibitor has been approved by the FDA for treating severe AA in adults and adolescents. Ruxolitinib and tofacitinib significantly increased hair shaft length, indicating their potential in promoting hair growth [20,21]. Clinical trials have also shown that baricitinib can lead to substantial hair regrowth, with some patients achieving at least 50% improvement compared to placebo [22]. Deuruxolitinib, a selective JAK1/JAK2 inhibitor, was recently FDA-approved for treating severe AA in adults [23,24]. Significant scalp hair regrowth was observed in approximately 65–70% of patients treated with deuruxolitinib, with improvements in hair thickness, coverage, and density. Unlike JAK1, which is broadly expressed in many tissues, JAK3 expression is largely restricted to hematopoietic cells. Of interest, selective inhibiting JAK3 signaling by ritlecitinib is sufficient to prevent and reverse AA [25]. 65–70% of patients in clinical trials achieved 50% or more scalp hair regrowth, and improvements in eyebrows and eyelashes were also noted [26,27]. In addition, many companies are developing JAK inhibitors as treatments for AA, as outlined in Table 2.

Although JAK inhibitors have proven beneficial in AA, they are not FDA-approved for AGA and can be costly. Known risks include heightened susceptibility to infections, thrombosis, and cardiovascular events, especially with prolonged use. For example, JAK inhibitors have been associated with an increased risk of deep vein thrombosis and pulmonary embolism, particularly in high-risk patients [31]. Long-term use may slightly elevate the risk of heart attacks and strokes, especially in older patients with pre-existing cardiovascular conditions [32]. Moreover, once patients discontinue therapy, previously regrown hair may be lost again.

## 3. Emerging Therapeutic Targets for Alopecia Treatment

Recent progress in understanding inflammatory processes, hormonal imbalances, stem cell dysfunction, and immune dysregulation has driven the discovery of novel molecular targets for treating various types of alopecia [33,34,35,36,37,38,39]. Table 3 outlines emerging targets such as AR and Wnt/β-catenin pathways, which play pivotal roles in hair follicle cycling, immune regulation, and stem cell function.

### 3.1. AR Inhibitors

AR inhibitors (ARIs) are pivotal in treating androgen-dependent conditions, notably prostate cancer and AGA [56,57]. Both systemic and topical ARIs are in development to enhance efficacy and mitigate resistance. Spironolactone, a potassium-sparing diuretic with anti-androgenic properties, is commonly used off-label for AGA and hirsutism [58,59]. It primarily functions by inhibiting AR, thus reducing androgenic signaling in HFs.

Clascoterone (Breezula^®^) is a topical AR inhibitor developed by Cassiopea, (San Diego, CA, USA) originally approved as Winlevi^®^ for acne and currently in trials for AGA [42,60]. Kintor Pharmaceuticals’ pyrilutamide, a nonsteroidal topical ARI, has shown promising preliminary results, including significant increases in hair count [43,61]. OliX Pharmaceuticals is investigating OLX72021, an RNA interference-based therapy targeting AR for AGA [46,62]. This approach aims to reduce AR expression, potentially mitigating hair loss. These developments underscore the growing potential of ARIs in hair loss treatment.

### 3.2. Wnt Activators

The Wnt/β-catenin pathway is integral to hair follicle development and regeneration, making it a prime target for AGA therapies [48]. Pharmaceutical companies are actively pursuing Wnt activators to promote hair growth.

Biosplice Therapeutics (formerly Samumed) has developed SM04554, a small-molecule Wnt modulator [63]. JW Pharmaceutical has developed JW0061 (GFRA1 agonist), a first-in-class drug candidate that promotes hair follicle proliferation and hair regeneration by activating the Wnt signaling pathway in skin and hair follicle stem cells [64,65]. KY19382, a novel CXXC5–Dvl interaction inhibitor, fosters Wnt/β-catenin signaling, thereby enhancing hair regrowth and wound-induced hair neogenesis [49].

### 3.3. Thyroid Receptor

Thyroid hormones play a crucial role in regulating metabolism, development, and tissue homeostasis, including hair follicle function [66]. Mice lacking TRα1 and TRβ (the main thyroid hormone binding isoforms) display impaired hair cycling associated to a decrease in follicular hair cell proliferation. In addition, TRα1/TRβ-deficient mice developed alopecia after serial depilation [67]. Thyroid hormone signaling is an important determinant of the mobilization of stem cells out of their niche in the hair bulge [68]. Therefore, research is exploring TRβ-selective agonists to fine-tune thyroid hormone effects while minimizing hair loss [69].

TDM-105795 is a topical small molecule drug candidate developed by Technoderma Medicines for the treatment of AGA. As a potent thyromimetic, it offers potential advantages in efficacy and safety over existing treatments [50,70]. A Phase 2a clinical trial involving mild to moderate AGA proved efficacy of TDM-105795.

### 3.4. Prostaglandin Derivatives

Prostaglandin (PG) derivatives have been found to influence hair growth in different ways. Specifically, prostaglandin F2α (PGF2α) derivatives promote hair growth, whereas prostaglandin D2 (PGD2) has been shown to inhibit it [71]. For example, PGD2 levels were approximately three times higher in balding scalp areas compared to non-balding regions in men with AGA [51]. Blocking PGD2 activity or its receptor may help prevent hair follicle miniaturization and promote hair regrowth.

Latanoprost and Bimatoprost, originally developed to reduce intraocular pressure in glaucoma treatment, were observed to induce hair growth as a side effect [72,73]. These compounds stimulate follicular cell proliferation and extend the anagen phase of the hair cycle, leading to increased hair density and length. DLQ01 is a topical prostaglandin F2α analog developed by Dermaliq Therapeutics for the treatment of AGA, and Dermaliq announced positive results from a Phase 1b/2a clinical trial. DLQ01 treatment resulted in a 12.3% increase in hair counts from baseline, and 83% of subjects treated with DLQ01 experienced positive hair growth.

### 3.5. Lactate Dehydrogenase (LDH)

Recent research has highlighted the role of lactate dehydrogenase (LDH) in hair follicle stem cell (HFSC) activation, offering promising avenues for developing novel hair loss treatments [74,75,76]. HFSCs utilize glycolytic metabolism, producing significant amounts of lactate. This lactate production is crucial for HFSC activation, as deleting the enzyme LDH in these cells prevented their activation. Conversely, increasing lactate production accelerated HFSC activation and the hair cycle [77].

Pelage Pharmaceuticals is developing a topical small molecule drug that targets this metabolic pathway. Their approach aims to activate dormant HFSCs by modulating lactate production, thereby stimulating hair growth. Pelage’s treatment is designed to be non-invasive and suitable for all genders, and hair types. They have initiated a Phase 2a clinical trial to evaluate the safety and efficacy of their lead compound, PP405, in individuals with AGA [52].

### 3.6. PDE4 Inhibitors

Phosphodiesterase 4 (PDE4) inhibitors are being explored as potential treatments for AA [78]. PDE4 is an enzyme that modulates inflammatory pathways, and its inhibition can reduce inflammation. Elevated PDE4 expression has been observed in AA, suggesting that PDE4 inhibition could reduce inflammation-associated hair loss [79].

Apremilast is a well-known PDE4 inhibitor (CC-10004), and recent studies have investigated its efficacy in promoting hair regrowth in patients with AA [53]. For instance, a Japanese patient with AA showed significant hair regrowth after 14 weeks of treatment with apremilast. Apremilast leads to the downregulation of inflammatory cytokines such as TNF-α, IL-17, and IFN-γ, which play a significant role in autoimmune diseases.

### 3.7. RIPK1 Inhibitor

Necroptosis, a programmed form of inflammatory cell death, has been increasingly recognized as a contributing factor to hair follicle dysfunction and hair loss [80]. By inhibiting RIPK1, RIPK3, or MLKL, necroptosis-related damage may be reduced. Necrostatin-1s (Nec1-s) has shown protective effects on human outer root sheath cells and improved hair regrowth in mouse models by limiting follicular inflammation [54].

## 4. New Therapeutic Modalities for Alopecia Treatment

Beyond small-molecule inhibitors, novel therapeutic modalities such as biologics and stem cell-based therapies aim to address hair loss more precisely, potentially improving upon traditional interventions. Beyond efficacy, future efforts must emphasize long-term safety and personalized approaches. The integration of antibody and cell-based therapies into mainstream clinical practice will be essential for meeting the unmet needs of patients.

### 4.1. Antibody Therapy

Antibody-based treatments have emerged as a targeted approach for AA, a condition in which existing immunosuppressants often yield variable results [81,82]. Antibodies are rarely developed for AGA therapy due to the requirement to be responsive to androgens and their receptors surrounding the hair follicle [83]. However, direct injection into the areas of hair loss is also feasible, potentially enhancing treatment efficacy while minimizing systemic side effects. Table 4 summarizes key antibody therapies under investigation.

Dupilumab, initially approved for atopic dermatitis, is under Phase 2 evaluation for AA [81]. Dupilumab is a humanized monoclonal antibody against IL-4Rα that downregulates TH2 response [91]. Clinical data suggested that subcutaneous injection of dupilumab every week slows AA progression, particularly in patients with atopic backgrounds [84]. Ustekinumab, which targets IL-12 and IL-23, has shown promise in case reports but remains off-label for AA [85]. Adalimumab, targeting TNF-α, yields inconsistent results and is not recommended as a standard AA treatment [86].

Prolactin (PRL) and its receptor (PRLR) play significant roles in hair follicle regulation and hair cycle stages. In humans and some mammals, PRL prolongs the telogen phase, potentially leading to reduced hair growth and increased shedding [92]. High PRL levels have been associated with delayed anagen initiation, contributing to hair thinning [93,94]. In addition, PRL can upregulate AR, potentially exacerbating hair follicle miniaturization in AGA [95]. Hope Medicine’s monoclonal antibody HMI-115 blocks PRLR and is in Phase 2 trials, with early reports suggesting promising efficacy [87].

CXCL12, also known as stromal cell-derived factor 1 (SDF-1), is a chemokine that plays a crucial role in tissue regeneration, immune cell recruitment, and stem cell homing [96,97,98]. Recent studies suggest that CXCL12 signaling may contribute to hair loss by influencing hair follicle cycling and the dermal microenvironment. For example, elevated CXCL12 levels in the dermis interfere with the transition from telogen to anagen, leading to prolonged hair loss [88]. In addition, secreted CXCL12 from dermal fibroblasts upregulated AR expression in DPCs to induce hair miniaturization in AGA [89]. Its interaction with immune cells, particularly through the CXCR4 receptor, may trigger chronic inflammation, which is associated with hair loss in AA [90]. Given the large molecular weight and long half-life of monoclonal antibodies, monthly or bimonthly subcutaneous injections of CXCL12 could be feasible for hair loss treatments.

### 4.2. Growth Factors and Platelet-Rich Plasma (PRP)

Therapeutic proteins have emerged as a promising approach to promoting hair growth by targeting various biological pathways involved in hair follicle regeneration. One notable therapeutic protein in hair growth research is basic fibroblast growth factor (bFGF), which extends the anagen phase of the hair cycle [99,100]. Additionally, vascular endothelial growth factor (VEGF) plays a crucial role in increasing blood supply to hair follicles, thereby promoting their activity and enhancing hair density [101,102]. Another important protein is insulin-like growth factor 1 (IGF-1), which has been shown to protect hair follicle cells from apoptosis [103,104]. Moreover, platelet-derived growth factor (PDGF) and keratinocyte growth factor (KGF, also known as FGF-7) contribute to hair follicle regeneration by supporting epithelial cell proliferation and follicle differentiation [105,106].

Platelet-rich plasma (PRP) has gained significant attention as a promising treatment for hair loss due to its regenerative properties (Table 5). PRP is derived from a patient’s own blood and contains a high concentration of growth factors (i.e., PDGF, VEGF, FGF, and IGF-1) that stimulate hair regeneration [107,108]. Typically, three initial treatments are administered monthly, followed by maintenance sessions every 3–6 months. Clinical trials suggested that PRP therapy effectively enhances hair density and thickness in women with hair loss, with a favorable safety profile [108]. However, the effects of PRP on hair density and thickness vary with dosage, injection duration, and ethnicity, indicating the need for tailored treatment protocols.

### 4.3. Botulinum Toxin

Botulinum toxin (Botox) has been studied for AGA, aiming to relax scalp muscles and potentially enhance blood flow to the follicles [109,117]. While preliminary studies show encouraging hair count increases in some participants, findings are not yet conclusive [118,119,120]. Botulinum toxin may be a promising therapeutic option for patients with various scalp conditions, but larger, randomized controlled trials are needed to better understand its efficacy and safety.

### 4.4. Cell Therapy

The development of cell therapy for hair loss treatment is gaining significant attention as an innovative approach utilizing DPCs, DSCs, and stem cells to stimulate hair growth [121]. Compared to conventional treatments, cell therapy may offer long-term efficacy with a single administration and fewer adverse effects [121]. Several companies are developing advanced therapies (Table 5).

The regulation of cell therapies for hair loss varies across countries, with key differences in how stem cell treatments are classified and approved. The U.S. and Europe have established frameworks for approving advanced therapies, with varying levels of clinical trial requirements and regulatory flexibility. Japan, China, and South Korea have more lenient regulatory practices, particularly for autologous stem cell treatments, as these therapies tend to be better tolerated and have fewer side effects. In these countries, regenerative medicine is regulated under the *Regenerative Medicine Safety Act* and the *Cell Therapy Act*, which allow for accelerated approval of cell therapies for hair loss.

Tissue grafting is a surgical procedure in which tissue is transferred from one area of the body to another. Plastic surgeons and dermatologists have reliably used scalp tissue grafting to improve and enhance hair regeneration. A simple procedure was developed to isolate epidermal stem cells (EpSCs) from scalp tissue and inject them into areas of hair loss to promote hair growth. Automated devices for producing stromal vascular fraction from lipoaspirated fat or EpSCs from scalp tissue are commercially available in clinics and have shown effectiveness in hair regeneration [122,123].

Shiseido has been actively involved in the development of hair regenerative medicine, focusing on the use of DSCs. Since initiating this research in 2016, Shiseido has collaborated with institutions such as Tokyo Medical University Hospital and Toho University Ohashi Medical Center to conduct clinical studies verifying the safety and efficacy of their proprietary cell processing product, S-DSC^®^. These studies have demonstrated positive results in treating male and female pattern baldness [110,111,124]. Shiseido’s S-DSC^®^ therapy represents a significant advancement in hair regenerative medicine. Shiseido began offering this treatment to patients in Japan on 1 July 2024. The procedure involves culturing autologous DSCs and injecting them into the scalp to promote hair growth.

DPCs, located at the base of hair follicles, play a pivotal role in regulating hair growth and cycling. Research has demonstrated that when DPCs are implanted between the epidermis and dermis, they can induce the formation of new hair follicles in hairless skin [125,126,127]. This finding underscores their potential in regenerative hair therapies. Currently, the treatment is in Phase 1/2a clinical trials for both male and female AGA. DPCs can be expanded in vitro while still maintaining their hair-inductive properties, while DSCs tend to lose their regenerative capacity more rapidly during in vitro expansion, limiting their potential for large-scale therapeutic applications [128]. As a result, DPC-based therapies hold greater promise for long-term and widespread use in regenerative medicine for hair loss treatment [112,121,127].

Adipose-derived stem cells (ADSCs) have emerged as a promising avenue for developing treatments for hair loss, particularly AGA [129,130]. These multipotent cells, harvested from adipose tissue, possess regenerative properties that can stimulate hair growth through various mechanisms [131,132,133,134]. Studies also have explored the use of ADSC-conditioned media (ADSC-CM) in hair regeneration therapies [135,136]. This approach involves applying the growth factors secreted by ADSCs to the scalp, promoting hair growth without the need for cell transplantation. Clinical trials have reported positive outcomes, indicating that cosmetics using ADSC-CM can effectively stimulate hair regeneration [137,138,139].

Stromal vascular fraction (SVF) is a component derived from adipose tissue and is gaining significant attention in regenerative medicine, including hair loss treatments [116,140,141]. SVF consists of a heterogeneous population of cells, including adipose-derived stem cells, endothelial cells, pericytes, fibroblasts, and immune cells, all of which contribute to tissue regeneration and repair [142]. Clinical studies have shown positive results, with patients experiencing increased hair density, thickness, and regrowth after SVF administration [114,115,116]. SVF helps improve the scalp’s microenvironment by increasing blood flow, reducing inflammation, and enhancing tissue repair [143].

### 4.5. Exosome for Hair Loss Treatments

Exosomes are tiny vesicles secreted by cells that contain various bioactive molecules, including proteins, lipids, and nucleic acids [144,145]. They play an important role in cell signaling and tissue regeneration and have recently garnered attention for their potential in treating hair loss. Current research indicates that exosomes are safe for use in hair loss treatment; however, data on their efficacy remain limited. A recent review highlighted that while preclinical studies have demonstrated positive effects of exosomes on hair growth, clinical data on their effectiveness and safety in humans are still insufficient [145]. Since exosome therapy is a relatively new field, further research is needed to establish its long-term effects and safety.

### 4.6. Innovations in Drug Delivery Systems for Hair Loss Treatments

Innovations in drug delivery systems for hair loss treatments are progressing rapidly. The development of novel finasteride and dutasteride formulations has generated considerable interest due to their potential for improved long-term safety and efficacy. For example, topical formulations aim to minimize systemic side effects by delivering the drug directly to the scalp, thereby reducing the risk of sexual dysfunction [146]. In addition, nanoparticle-based delivery systems enhance drug stability and therapeutic efficacy, while controlled-release mechanisms may improve patient compliance by reducing the frequency of application [147,148].

Lipid-based nanoparticles stabilize drugs by effectively encapsulating both hydrophilic and lipophilic substances. Nanostructured lipid carriers enhance drug stability, solubility, and sustained release, thereby improving drug penetration into the scalp. This delivery method may enhance the effectiveness and reduce the side effects of existing drugs such as minoxidil and finasteride [149].

Microneedles are used to deliver drugs directly into the scalp, enhancing absorption and maximizing treatment efficacy. Recently, they have been actively studied in the field of hair loss treatment as an alternative approach to overcome the limitations of oral medications and topical applications. For example, microneedles can effectively penetrate the stratum corneum of the scalp, delivering active ingredients such as minoxidil or finasteride [150,151,152]. Microneedle systems loaded with exosomes derived from stem cells and growth factors (e.g., bFGF, VEGF) have been developed to promote follicle regeneration [153,154,155].

3D-printed scaffolds also represent an innovative approach that is gaining attention in the field of hair restoration. This technique involves the use of 3D printing technology to create customized scaffolds that can be implanted into the scalp to promote hair growth. The scaffolds are designed to match the individual’s scalp condition and can be loaded with drugs for gradual release, thereby maximizing therapeutic effects.

## 5. Conclusions

Advances in molecular biology, regenerative medicine, and precision therapeutics are redefining the landscape of hair loss drug development. While current mainstays such as minoxidil and finasteride have provided some benefit, their limitations underscore the critical need for more targeted and effective therapies. Emerging modalities, including JAK inhibitors, Wnt activators, and gene-editing approaches, offer the potential for transformative advancements in hair restoration.

Antibody therapy and cell therapy, in particular, represent highly promising avenues for the development of next-generation hair loss treatments. Antibody-based therapies can provide precise targeting of key molecular pathways implicated in hair follicle cycling, offering enhanced specificity and minimized off-target effects. Meanwhile, cell-based approaches, including follicular stem cell transplantation and ex vivo expanded DPCs, have the potential to directly regenerate hair follicles and restore hair growth. As these therapies advance, their integration into the therapeutic pipeline will be critical to overcoming the limitations of current treatments.

One of the major challenges in this field is patient heterogeneity. Genetic variations, immune profiles, and hormonal influences can significantly affect treatment outcomes, making a one-size-fits-all approach inadequate. Personalized medicine, supported by AI-driven drug discovery and biomarker identification, holds promise for tailoring interventions to individual patients, thereby optimizing therapeutic efficacy. Additionally, innovations in drug delivery technologies—such as microneedles, nanoparticles, and 3D-printed scaffolds—could further enhance treatment outcomes by ensuring precise and sustained release of active compounds.

Combination therapies that leverage multiple mechanisms, including immunomodulation, follicle stem cell activation, and angiogenesis, are likely to surpass the efficacy of single-agent strategies. As gene- and cell-based therapies advance toward clinical application, ethical and regulatory considerations will play a pivotal role in ensuring their safe and equitable use.

Ultimately, the future of hair loss therapeutics will depend on interdisciplinary collaboration among dermatologists, bioengineers, pharmaceutical scientists, and AI experts. By integrating diverse expertise and prioritizing the development of antibody- and cell-based therapies, the field can accelerate the creation of innovative solutions to address the unmet needs of patients with hair loss disorders.

## Figures and Tables

**Figure 1 ijms-26-03461-f001:**
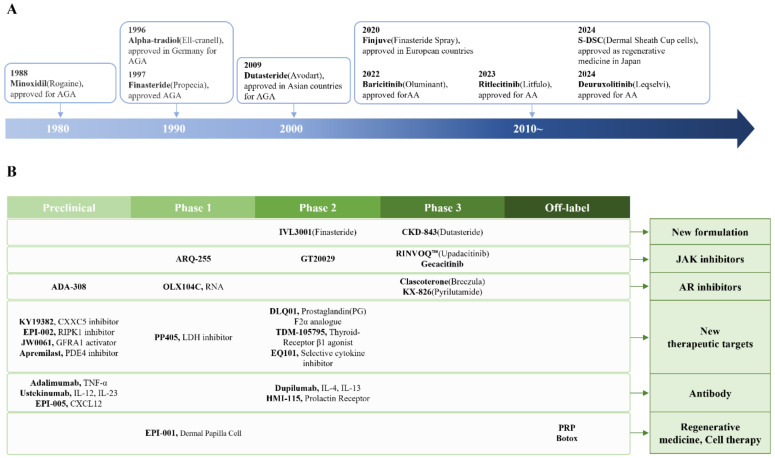
Approved drugs (**A**) and new drug candidates (**B**) for hair loss disorders.

**Table 1 ijms-26-03461-t001:** Approved drugs for hair loss treatment.

Brand Name	General Name	Mechanism of Action	Advantages and Challenges
**Rogaine**	Minoxidil	Stimulates proliferation of dermal papilla cells Inhibits collagen synthesis Promotes VEGF expression Opens KATP channels and induces vasodilation	Generally safe with mild and temporary side effects Suitable for both men and women, though 2% is typically recommended for women
**Ell-Cranell**	a-tradiol	Reduces DHT levels Promotes synthesis of 17β-estradiol	Minimal hormonal impact Less effective for hair regrowth when used alone
**Propecia**	Finasteride	Inhibits type II 5α-reductase Inactivates androgen receptors Inhibits fibrosis	Widely used for male pattern baldness Simple once-daily oral intake (1 mg) Possible sexual side effects in ~1–2% of users
**Avodart**	Dutasteride	Inhibits type I and II 5α-reductase Inactivates androgen receptors Inhibits fibrosis	Not FDA-approved for hair loss in the U.S. Approved in Japan and South Korea Long half-lif
**Finjuve**	Finasteride	Inhibits 5α-reductase Inactivates androgen receptors Inhibits fibrosi	Topical administration
**Olumiant**	Baricitinib	Inhibition of JAK1/2 Reduces T cell activation and cytokine production Restores immune privilege of the hair follicle	Significant regrowth in severe alopecia areata (≥50% scalp hair loss) Rapid onset (noticeable regrowth in 3–6 months)
**Litfulo**	Ritlecitinib	Selectively inhibits JAK3 and TEC kinases Blocks autoimmune attack on hair follicles	FDA-approved for patients aged 12 and older May reduce side effects by avoiding broader immune suppression
**Leqselvi**	Deuruxolitinib	Inhibits JAK1/2	Recently approved Well tolerated Effective for severe cases

**Table 2 ijms-26-03461-t002:** Emerging therapeutic target and its development.

Target/MoA	Product/Pipeline	Company	Stage of Development	Route of Administration	Reference
**JAK 1**	RINVOQ™ (Upadacitinib)	ABBEVI (North Chicago, IL, USA)	P3	Oral	[28]
**JAK**	Gecacitinib (Jaktinib)	Suzhou Zelgen Biopharmaceuticals (Suzhou, China)	P3	Oral	[29]
**JAK 1**	ARQ-255	Arcutis Biotherapeutics (Westlake Village, CA, USA)	P1b	Oral	[30]

**Table 3 ijms-26-03461-t003:** Emerging therapeutic target and its development.

Target/MoA	Product/Pipeline	Company	Stage of Development	Route of Administration	Reference
**AR Antagonist**	Breezula (Clascoterone)	Cosmo Pharmaceuticals (Lainate, Milan, Italy)	P3	Topical	[40,41,42]
**AR Antagonist**	KX-826 (Pyrilutamide)	Kintor Pharmaceuticals (Suzhou, China)	P3	Topical	[43,44]
**AR Degrader**	GT20029	Kintor Pharmaceuticals	P2	Topical	[45]
**AR RNAi**	OLX104C	OliX Pharmaceuticals (Suwon, Gyeonggi, Republic of Korea)	P1	Intradermal injection	[46]
**AR Antagonist**	ADA-308	Aranda Pharma (Kuopio, Finland)	Preclinical	Topical	[47]
**GFRA1 agonist (Wnt Activator)**	JW0061	JW Pharmaceutical (Gwacheon, Gyeonggi, Republic of Korea)	Preclinical	Topical	[48]
**CXXC5 Inhibitor (Wnt Activator)**	KY19382	CK Regeon (Seoul, Republic of Korea)	Preclinical	Topical	[49]
**Thyroid-Receptor β1 Agonist**	TDM-105795	Technoderma Medicines (Chengdu, China)	P2	Topical	[50]
**Prostaglandin F2α**	DLQ01	Dermaliq Therapeutics (Wilmington, DE, USA)	P2a	Topical	[51]
**lactate dehydrogenase (LDH)**	PP405	Pelage Pharmaceuticals (Los Angeles, CA, USA)	Phase 1	Topical	[52]
**PDE4 inhibitor**	Apremilast (CC-10004)	Celegene/BMS (Summit, NJ, USA)	Case report	Oral	[53]
**RIPK1 inhibitor**	Necrostatin-1s	Epi Biotech (Incheon, Republic of Korea)	Preclinical	Topical	[54]
**Selective cytokine inhibitor**	EQ101	Equillium (La Jolla, CA, USA)	P2	Intravenous injection	[55]

**Table 4 ijms-26-03461-t004:** Antibody therapy for hair loss.

Target Gene	Product/Pipelines	Company	Type of Alopecia	Stage of Development	Reference
**IL-4 and IL-13**	Dupilumab	Sanofi (Paris, France)	AA	P2	[84]
**IL-12 and IL-23**	Ustekinumab	Janssen Biotech (Horsham, PA, USA)	AA	Case report	[85]
**TNF-a**	Adalimumab	AbbVie (North Chicago, IL, USA)	AA	Case report	[86]
**Prolactin receptor**	HMI-115	Hope Medicine (Pudong, Shanghai, China)	AGA	P2	[87]
**CXCL12**	EPI-005	Epi Biotech	AGA, AA	preclinical	[88,89,90]

**Table 5 ijms-26-03461-t005:** Regenerative medicine for hair loss treatment.

Product/Pipelines	Company	Stage of development	Reference
**PRP**	-	Marketed	[107,108]
**Botulinum toxin/Nabota**	Daewoong Pharmaceutics (Seoul, Republic of Korea)	Marketed	[109]
**Dermal sheath cup cells**	Shisheido (Tokyo, Japan)	Marketed	[110,111]
**Dermal papilla cells**	Epi Biotech	P1/2a	[112,113]
**Stromal vascular fraction**	-	Marketed	[114,115,116]

## Data Availability

Not applicable.

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
