# Peer review of "Recent Advances in Drug Development for Hair Loss"

_ijms, 2025, doi:10.3390/ijms26083461_

Round 1
Reviewer 1 Report
Comments and Suggestions for Authors
The authors have explored various metabolic pathways associated with hair loss and have cataloged both approved treatment plans and promising approaches in different phases of clinical trials. Nonetheless, a more comprehensive discussion on the benefits and challenges of medications targeting each metabolic pathway is necessary. Please consider the following comments for further manuscript revision.
- The authors provided commendable references for each metabolic pathway associated with hair growth. However, there is a lack of detailed information on the specific improvements observed, such as changes in hair diameter, hair cross-sectional area, hair density, regrowth percentage, duration of hair growth, as well as aspects related to safety, efficacy, and bioavailability.
- Elucidate why dutasteride exhibits greater efficacy than finasteride, despite sharing the same mechanism of action as 5 a-reductase inhibitors.
- The discussion regarding the side effects of JAK inhibitors lacks clarity and supporting citations.
- The authors have identified medications targeting multiple metabolic pathways for androgenetic alopecia (AGA) and alopecia areata (AA). A comprehensive table listing these medications, specifying their respective metabolic pathways, and discussing the advantages and challenges associated with each pathway would be beneficial.
- The reference list appears incomplete, as citations are only visible up to number 68, despite the manuscript indicating a total of 138 citations.
Reviewer 2 Report
Comments and Suggestions for Authors
The paper is significant, informative and well written. However, some sections need significant amount of additional disscussion and information, thus I would reccomend major revision. Details could be found in separate comments.
Line 51 Section 2 “Current Hair Loss Treatments” could be rephrased “Current approved Hair Loss Treatments” if it refers only to approved drugs.
In line 64 provide some data on the efficacy of minoxidil in percentages from most recent trials with minoxidil, e.g. study by Rundegren showed that the reported affected area had become smaller in 561 of the 904 eligible subjects (62.0%), was unchanged in 317 subjects (35.1%), and had become larger in 26 subjects (2.9%).( https://www.jaad.org/article/S0190-9622(03)03692-2/fulltext) This is just an example, provide studies from past 10 years.
The same should be done for 2.2. 5 a-reductase inhibitors, provide some data on the efficacy of Finasteride and dutasteride in percentages from most recent trials.
In the Section Cell therapy, provide some data about the usage of these methods in world countries,in percantages. Also, you could mention limitations in usage of cell based products in regulatory terms that is different in Europe,USA and Asia. Also, emphasize the significance of the autologous treatments in terms of better tolerance and less side effects.
You are mentioning follicular stem cell transplantation for the first time in conclusion but there is no discussion about this approach .Add this in the section Cell therapy.
Also, in the same section Cell therapy provide data and discussion on the use of exosomes as there are trials suggesting they may fuel hair growth (doi: 10.2147/CCID.S465963; DOI: 10.1097/DSS.0000000000004480 ; https://doi.org/10.1016/j.jddst.2023.105126; https://doi.org/10.1007/s00266-024-04332-3). They are very important in latest developments of cell based products in Hair Loss Treatments.
Drug delivery is another important point in success of these therapies. In the Conclusion the authors state “Additionally, innovations in drug delivery technologies-such as microneedles, nanoparticles, and 3D- printed scaffolds-could further enhance treatment outcomes by ensuring precise and sustained release of active compounds.” You mention this only in the conclusions. Please provide more broad Disscussion in separate section 4.5. called “Innovations in drug delivery systems for Hair Loss Treatments” as this is an important aspect of hair loss treatments.
Round 2
Reviewer 1 Report
Comments and Suggestions for Authors
My comments were addressed. No further comments from me.
Reviewer 2 Report
Comments and Suggestions for Authors
The authors have incorporated all suggestions and significanlty improved the quality of the manuscript.The paper is acceptable in current form.
Minor correction: Provide reference for the last paragraph in the Disscussion:3D-printed scaffolds also represent an innovative approach that is gaining attention
in the field of hair restoration.